# Earth Science Education as a Key Component of Education for Sustainability

**Clara Vasconcelos** [1,*] **and Nir Orion** [2,*]

1    Interdisciplinary Centre of Marine and Environmental Research (CIIMAR), Science Teaching Unit (UEC) & Department of Geosciences, Environment and Spatial Planning (DGAOT), Faculty of Sciences (FCUP), University of Porto, 4169-007 Porto, Portugal
2    Science Teaching Department, Weizmann Institute of Science, Rehovot 7610001, Israel
*    Correspondence: csvascon@fc.up.pt (C.V.); nir.orion@weizmann.ac.il (N.O.);
      Tel.: +97-289-344-043 (C.V.); +220-402-462 (N.O.)

**Abstract:** Environmental insight has emerged as a new scientific concept which incorporates the understanding that the Earth is made up of interworking subsystems and the acceptance that humans must act in harmony with the Earth's dynamic balanced cycle. This Earth system competency represents the highest level of knowing and understanding in the geosciences community. Humans have an important role as participative beings in the Earth's subsystems, and they must therefore acknowledge that life on Earth depends on a geoethically responsible management of the Earth system. Yet, the world is far from achieving sustainable development, making the role of the Earth science education in promoting education for sustainability even more relevant. The Earth system approach to education is designed to be an effective learning tool for the development of the innovative concept of environmental insight. Through a holistic view of planet Earth, students realize that humans have the ability to enjoy a sustainable life on our planet while minimising detrimental environmental impacts. There is growing evidence that citizens value science and need to be informed about Earth system problems such as climate change, resource efficiency, pandemics, sustainable use of water resources, and how to protect bio-geodiversity. By moving away from both traditional practices and traditional perceptions, environmental insight and geoethics will lead towards an education for sustainability that provides the citizens of Earth with the tools they need to address the full complexity of its urgent environmental concerns.

**Keywords:** Anthropocene; Covid-19 pandemic; Earth system governance; education for sustainability; environment

## 1. Introduction

Since September 2015, the sustainable development goals of the United Nations have been adopted in all fields of knowledge. From science to education, the focus on sustainability on Earth has come a long way, teaching and communicating to all citizens that Earth is a dynamic system. Nevertheless, the impact of humans on our planet has grown dramatically over the last centuries, triggering major, and strongly negative, impacts on the Earth system. Human behaviour is threatening to overwhelm the sustainability of the Earth subsystems. This costly impact is endangering the dynamic of the Earth system with potentially serious consequences. Life on Earth may become unbearable if we do not greatly enhance the awareness of citizens regarding a holistic view of the Earth system and increase geoethical behaviours.

The Earth sciences study and explore the Earth system and are therefore highly relevant to understanding and addressing central environmental issues. They incorporate such topics as the mutual influence among natural systems (human involvement excluded), the influence of human intervention on the earth system, the ability to forecast disastrous natural phenomena, the use of physical environment to produce energy, the sustainable

development of natural resources, and global changes in climate [1]. In doing so, the Earth sciences could play a central role in facilitating the attainment of what Biermann et al. have referred to as an "Earth system governance" [2].

"Earth system governance" refers to the process of defining and developing socio-economic systems that will prevent drastic Earth system disruptions [2,3]. While consumption, inequality, and population have increased extremely quickly, processes such as the extraction of resources from the system and the return of waste and pollution, climate change, and the unregulated mining of raw materials are also simultaneously altering land cover, fragmenting ecosystems, and reducing bio and geodiversity [4]. As a result, if we do not change our day-to-day activities, humanity may find that its environment is no longer fit to live in.

To instil sustainability in our daily routines, we must go beyond the common actions. An effective Earth system governance must reflect on Earth system signals and answer to the imperative call for action. However, how can we achieve this ambitious aim? Does Earth science education have the potential to change human behaviour? How can Earth science educators promote such an attitudinal change?

In this paper, the authors present a review of the literature regarding the Earth system education approach and the development of environmental insight. It then addresses the high potential of Earth science education as a key component of education for sustainability. It concludes by arguing for the ultimate goal of promoting a change in education, updating it to our times and to the threats that Earth is facing. From climate change to pandemics, a holistic view of planet Earth can make the difference between just surviving or living as a human community.

## 2. Education for Sustainability

With globalization at the core of the contemporary economic and social dynamics, knowledge (and consequently education) stands as a progressively powerful determinant of comparative advantage.

### 2.1. From Environment Education to Education for Sustainability

The concept of sustainable development was popularized in 1987 in response to questions raised about the need to redefine the notion of "development" with regard to the persistent degradation of environmental quality [5], in conjunction with social and economic disparities all over the world. This concept was clarified and defined in the "Our Common Future" report (known also as Brundtland report) as development capable of satisfying the needs of the present generations without putting the needs of future generations at risk [6]. Environmental conscience and concern were recognized as vectors for the development and the evolution of the human civilization, and environmental issues stopped being exclusively technical, instead becoming intertwined with sociocultural and educational aspects. According to the Our Common Future report, sustainable development requires meeting the basic needs of all and extending to all the opportunity to satisfy their aspirations for a better life. In other words, sustainability refers to development efforts and practices that target a balance between environmental, economic, and social needs of the present as well as future generations [7].

In the Johannesburg Earth Summit in 2002, the United Nations declared 2005–2014 as the decade for sustainable development, changing the paradigm of environmental education by equating it with education for sustainable development. This declaration generated some disagreement among educators regarding the relationship between "education for sustainability" and "environmental education" and whether the two were indeed interchangeable [8]. The decade of education for sustainable development resulted in an international shift from discourse on environmental education to education for sustainable development (or, as it is often called, education for sustainability), but the question remains whether the change in discourse and language was accompanied by a real change in educational practice [9]. The disagreements over the designation of that educational

dimension led Freitas [10] to consider that both environmental education and education for sustainability could coexist as close relatives.

By September 2015, 195 nations had agreed to join forces to develop Agenda 2030—a plan of action with 17 sustainable development goals (SDGs) and 169 targets. These goals and targets were integrated, indivisible and interlinked, thus their implementation requires an in-depth discussion and a comprehensive understanding of the inter-connection between them. Moreover, to accomplish the United Nations' agenda, governments, media, businesses, stakeholders, institutions of education, and non-profit organizations (NGOs) will have to work together and build a robust and fruitful partnership by the year 2030.

Formal sustainability education programs devoted to including rigorous systems of thinking, social interactions, and green principles have emerged in the last decade, and its success is generating initiatives to change the curricula [11]. From developing competences in primary pupils to engage higher education students, sustainability projects also demonstrate that integral approaches are urgently needed [12]. Research points out that teaching methods providing a good introduction, supportive guidelines, and including active participation and interactivity are the ones that integrate the most effective strategies for developing learners' understanding, thinking, and ability to act for sustainability [13,14]. Unfortunately, the slow adaptation of the educational systems to the education for sustainability paradigm has created a huge gap between Agenda 2030's intentions and the accomplishment of its goals. This gap constitutes part of the essence of a worldwide educational problem. The social changes that the present calls for require teachers in general—and especially science teachers—to acquire new theoretical knowledge and internalize a new set of values and attitudes [15]. Science communication is also a powerful tool with which to influence societal attitudes towards planet Earth. Unfortunately, social movements against science have risen over the years in areas such as climate change, nuclear energy. and mining. As we live through the COVID-19 pandemic, studies suggest that higher levels of distrust in science correspond to greater unwillingness to engage in social distancing measures [16–18]. When working against a wave of conspiracy theories surrounding science, clear, succinct, and efficient science communication is crucial to addressing the problem of climate change or any other similarly controversial topics [19,20].

### 2.2. Earth Sciences and Education for Sustainability

Literature tells us that educators who can implement new educational resources and strategies can lead to good practices in the field of sustainability and promote hope and action among their students to inspire them to become social innovators [21–23]. Achieving the United Nations' SDGs calls for social innovation, that is, for changes in society's attitude and behaviours in order to ensure the sustainability of life on Earth, especially when anthropogenic changes are threatening our planet [24–26]. Environmentalism is a well-known social movement, and much has already been invested in promoting social change via a huge range of environmental behaviours on multiple scales (ranging from energy conservation and the adoption of green technology to household composting and community gardening in urban areas) [24,27]. The gap between pro-environmental attitudes to actual changes in behaviour has proven difficult to bridge, with reports of high levels of pro-environmental attitudes not necessarily corresponding to equally high levels of pro-environmental behaviours. Nevertheless, changes in attitudes are important, and this is where education for sustainability may also play an important role [28–30]. Orion and Libarkin suggested that the focus of the traditional environmental movement on the development of environmental awareness has failed to change the environmental behaviour of citizens worldwide and pointed to the inherent potential of Earth sciences to better address the challenge of changing environmental behaviour [31].

The Earth sciences focus on understanding how the Earth's subsystems function and interact [1]. Helping students recognize how processes that operate on planet Earth interact to generate physical and biological diversity over vast spatial and temporal scales is a

quality unique to the Earth sciences [31]. The dynamic balance of Earth's subsystems defines how our planet deals with unexpected agents that cause natural disruptions of the balance between them. Understanding how this dynamic works is of capital relevance to understanding Earth's sustainability and to directing our behaviours towards SDGs, which places the geosciences as one of the major areas that can be used by teachers in conceptualizing sustainability [32–35].

However, the understanding of Earth as a system should involve the development of systems thinking. "Systems thinking" is a term originally used to indicate a holistic approach through which to account for the dynamic interdependencies among parts in a whole, or, in other words, for seeing a whole as a sum of its parts [36]. This competency of thinking has been applied to many fields and disciplines, including the geosciences, where helping students to develop system thinking has been defined as one of the discipline's foremost challenges [37].

Reflecting on a possible reorientation in geoscience curricula directed at education for sustainability, both at a conceptual content level and in terms of teaching methodologies and strategies, might be an efficient answer to the current challenges of geosciences teaching. The way in which teachers and students conceptualize sustainability will influence the ways in which they anchor it into their processes of teaching and learning [38,39].

### 3. Earth System, Environmental Insight, and Earth Science Sustainability-Based Education

The educational Earth system approach is based on the construction of knowledge by learners through the mediation of the teacher and is therefore based on a close engagement of the learner in the learning process. The ultimate aim of Earth system education is the development of environmental insight. This competency comprises the ability to (a) recognize the interworking relationships between Earth subsystems and (b) to reflect on one's own role in system Earth so as to continually evaluate geoethical behaviours to preserve life on Earth. Orion [1] defined two main dimensions that students must develop to attain environmental insight: (i) the understanding that Earth system integrates interconnected subsystems (geosphere, hydrosphere, biosphere, and atmosphere) which exchange energy and materials; and (ii) the understanding that humans are a part of the Earth system and thus must act in harmony with its "laws" of cycling. The study of the interacting Earth systems—within the dimension of deep time and the large spatial scale of geological processes—will enable students to appreciate the realistic influence of humans on the Earth in deep time perception. This approach moves students away from the traditional altruistic approach to environmental awareness towards a more egocentric and geocentric perception [40,41].

Unfortunately, the traditional altruistic approach, which characterized the environmental movement for over half of a century, is probably one of the reasons for the slow change in the public's environmental behaviour. Many environmental NGOs have declared their altruistic approach to activism through names such as "Earth Watchers", "The Society for the Protection of Nature", and "Friends of the Earth". Moreover, the public has been exposed for decades to governmental campaigns to encourage environmental behaviours (recycling, saving energy, etc.) that were based on altruistic slogans such as "save the planet", "save the environment", and "the Earth is in our hands". The balance among the Earth systems is fragile and has been disrupted endless times over the billions of years of Earth's existence. However, the feedback mechanism of the Earth system allows subsystems to return to balance each time. This process leads to changes in all spheres, for example, in the biosphere, some species become extinct and new species develop. This is the essence of environmental insight—to understand the realistic role of humans on Earth. In other words, the Earth will survive and recover from the imbalance of the Earth's subsystems, but the human race may not survive. Thus, the meaning of education for sustainability is the understanding that the Earth system is sustainable and that the subsystems sustain each other. However, although the biosphere is sustainable for billions of

years, parts of it are not, and the imbalance that humanity is causing has put *us* at risk—not the Earth.

Levy and collaborators have shown that environmental behaviour is often driven by "egoistic" concerns rather than by altruistic views and motivations [41]. They claimed that their findings suggest that part of the failure of all these campaigns to establish a significant and sustainable change in the public's environmental behaviour (at least in several countries) may be related to their altruistic approach. Consequently, they suggested that, in order to convince citizens to change their environmental behaviour, educational programs and environmental campaigns should target humanity's egocentric nature.

The achievement of environmental insight evolves together with the higher-order cognitive competency of system thinking, the development of which requires the implementation of a holistic Earth system approach emphasizing the study of the cyclic pattern of the transformation of matter and energy among the subsystems [1]. According to the same author and some collaborators [42,43], the system thinking competency demands the understanding of a variety of scientific, technological, and social domains, thus there should be a close relationship between environmental insight and Earth system thinking. Therefore, both competencies need to be developed *in* and *for* an Earth science sustainability-based education.

New challenges driven by globalization and technological advancements have led to a growing international recognition that preparing students for success includes the development of higher order thinking and competencies. These 21st century competencies differ from traditional academic ones by not being primarily content knowledge-based. Both critical thinking and system thinking continue to be mentioned as cross-cutting key competencies for sustainable development [44]. However, the meaningful contribution of the Earth system approach and the environmental insight competency as key competencies for sustainable development is not yet being acknowledged [42].

The COVID-19 pandemic is an example of how environmental insight is critical to preserving life on Earth (Figure 1). Most recent pandemics have been zoonotic [45], originating in wildlife, and the emergence of infectious diseases seems to correlate with human population density and wildlife diversity, accelerated by anthropogenic changes such as deforestation, intensification of livestock production, and increased hunting and wildlife trade [46]. Preserving ecosystems and their endemic biodiversity is important when it comes to disease control, keeping possible pathogens inside those communities [46,47]. Climate change plays an important part in the emergence of infectious diseases, especially when it leads to floods, which, in turn, lead to an increased risk of transmission of water-borne diseases, such as cholera [46,48].

There are four main modes of infectious disease transmission caused by microorganisms, such as viruses and bacteria: contact (direct or indirect), airborne, common vehicle (water or food), and vector-borne (insects or vermin) [49]. These modes of transmission, along with the known consequences that climate change and the disruption of ecosystems have on the emergence of infectious diseases, highlight some of the different Earth subsystems that can be involved in the process. The SARS-CoV-2 (severe acute respiratory syndrome coronavirus 2), also of zoonotic origin, seems to be related to an interaction within the biosphere that led to the transmission of the virus from animals to humans, followed by its spreading through contact (direct and indirect) and through air, therefore linking it to the atmosphere. The lockdown imposed in most cities following the outbreak has led to changes in societal behaviours, which, in turn, has had consequences on all other Earth subsystems: the geosphere, with the reduction of coal and oil consumption; the hydrosphere, with reports in improvement of water quality; and, again, the atmosphere, with a reduction of air pollution (reduction of traffic and gas emissions) and an improvement in air quality [50]. However, not everything is positive; following the coronavirus outbreak, there has been a growing amount of unrecyclable medical waste (gloves, plastic hand sanitizers' bottles, and masks) contaminating the rivers and the oceans as well as an in landfills. The rise in unemployment numbers reflects the economic and the societal

problems that can stem from the pandemic [51], which in turn represents the potential cost of the current lack of sustainable development all over the world.

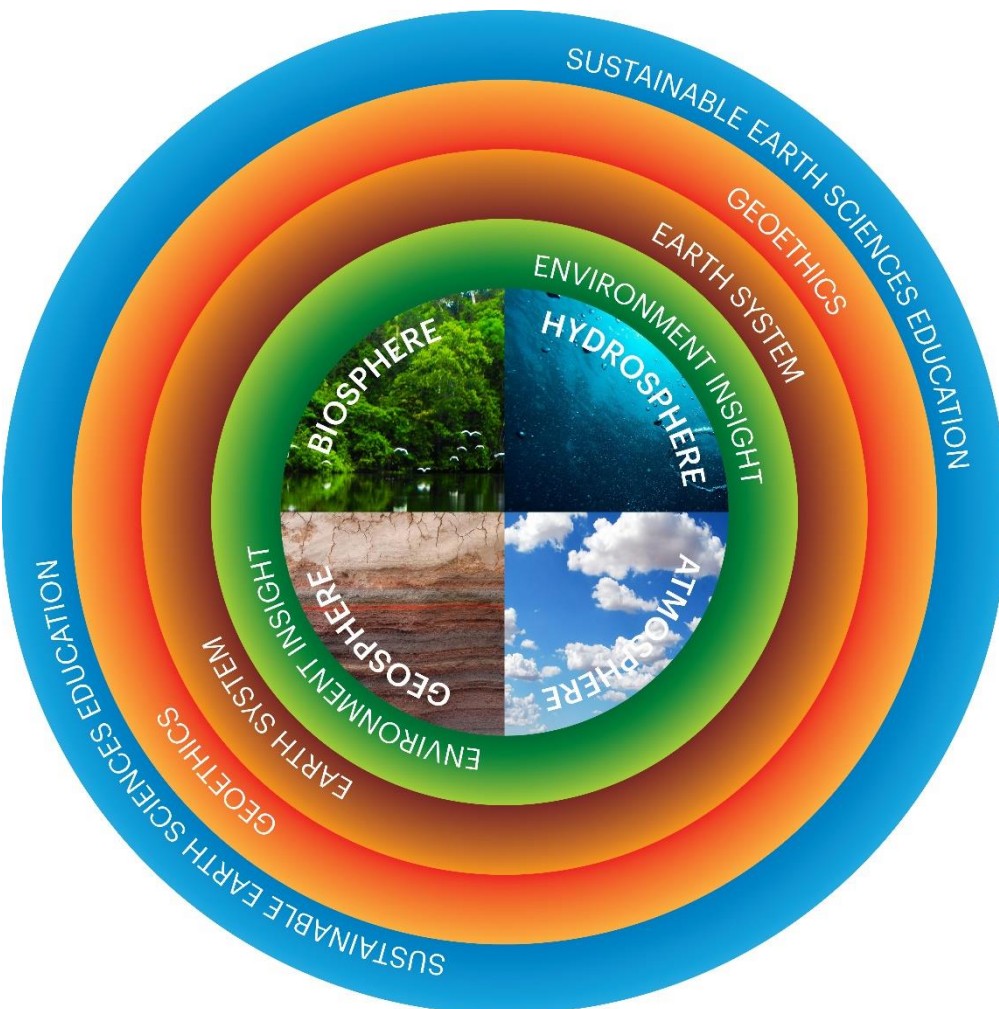

**Figure 1.** The environmental insight required to understand the case of COVID-19 outbreak.

The pandemic we are experiencing is the greatest proof of the holistic vision that we must have of the Earth system. As humans can only change behaviours after understanding the consequences of their actions, citizens need to have a holistic view of the Earth system to geoethically reflect upon their place on our planet. Environmental insight is a powerful competency with which to assist and guide citizens towards an awareness of the efforts needed to overcome Earth system disruption. The ultimate aim of the Earth system approach is the development of environmental insight [42].

## 4. Geoethics and Earth System Governance

There is an intricate relationship between environment, economic activity, growth, and social well-being. Respect for this relationship is a paramount concern for the future of humankind. However, this apparently straightforward idea is hard to pursue, especially since citizens (who act as social, economic, and political agents) are not aware of the consequences of their behaviours or of the alternatives at their disposal [52].

Unfortunately, there is a disturbing gap between the educational potential of Earth sciences and its low profile in schools, and—consequently—the public's Earth science literacy. Narrowing this gap requires holistic efforts on multiple fronts, one of which is in the hands of the geoscience community [40]. A deep change in the status of Earth science education in schools requires a deep change in the attitudes of geoscientists towards their

social responsibility. This responsibility is part of the professional ethics of a geoscientist, as announced in 2016, during the 35th International Geological Congress in Cape Town, South Africa, in the document known as the "Cape Town Statement on Geoethics" [53]. Geoethics can help people re-evaluate their behaviour and increase their awareness of alternative human activities. It is therefore imperative that it be taught in geoscience classes and extended to citizens through social innovation or outreach activities.

The introduction of an ethical dimension in (geo)science education is not exactly a new approach [54], but its prominence has increased in recent years. Having understood the importance of the human impact on nature, Antonio Stoppani (1824–1891) presented, in the early nineteenth century, the concept of "Anthropozoic". This concept was used to define the latest era in the designation of geological time in which human action began to be significant and decisive in the dynamic of the Earth system [55]. It anticipated the modern concept of Anthropocene [56,57] and is presently considered a preamble of the reflection which, in late nineteenth century, led to the emergence of geoethics [55,58,59].

Geoethics emerged as a new field of research in the early 1990s and is now a consolidated field of study with many publications and conferences devoted to the topic worldwide. Geoethics consists of research and reflection on the values that underpin appropriate behaviours and practices wherever human activities interact with the Earth system [60]. Geoethics covers many aspects of geosciences, from establishing clear and transparent professional codes of practice to global perspectives and governance around Earth system destruction [40]. The relative novelty of geoethics as a field of study makes disseminating geoethics knowledge a pressing task, not only amongst future geologists and geology teachers but also amongst students and citizens [4].

Some innovative resources for teaching scientific dilemmas that deal with geoethics implications were developed in 2020 as part of an international project undertaken by scientists and geoscience educators [61]. The resources refer to geoethics in water management, natural risks, geoheritage, and mining and were developed to be applied in higher education. The conceptual basis of this work was to contribute to a better understanding of scientific concepts and of the complexity of ethical issues. It demonstrated that democratic societies must address the challenges associated with the exponential growth of sustainability problems on Earth and that their resolution requires knowledge, expertise, and ethics. The eBook published in light of this research project describes the efforts currently being invested in building a socially responsible and ethically committed future geoscientific community. As noted by Peppoloni and Di Capua, " . . . beyond the social commitment of geo-scientists, it is the task of all society to undertake a general mobilization aimed at choosing its future and ensuring its safety, health and sustainability" [62] (p. 4).

Teaching geoethics, along with the promotion of environmental insight, will enable a shift towards a cleaner environment and a healthier and more equitable society. This will not only lead to a sustainable Earth science education (Figure 2) but also foster a sustainable Earth system governance.

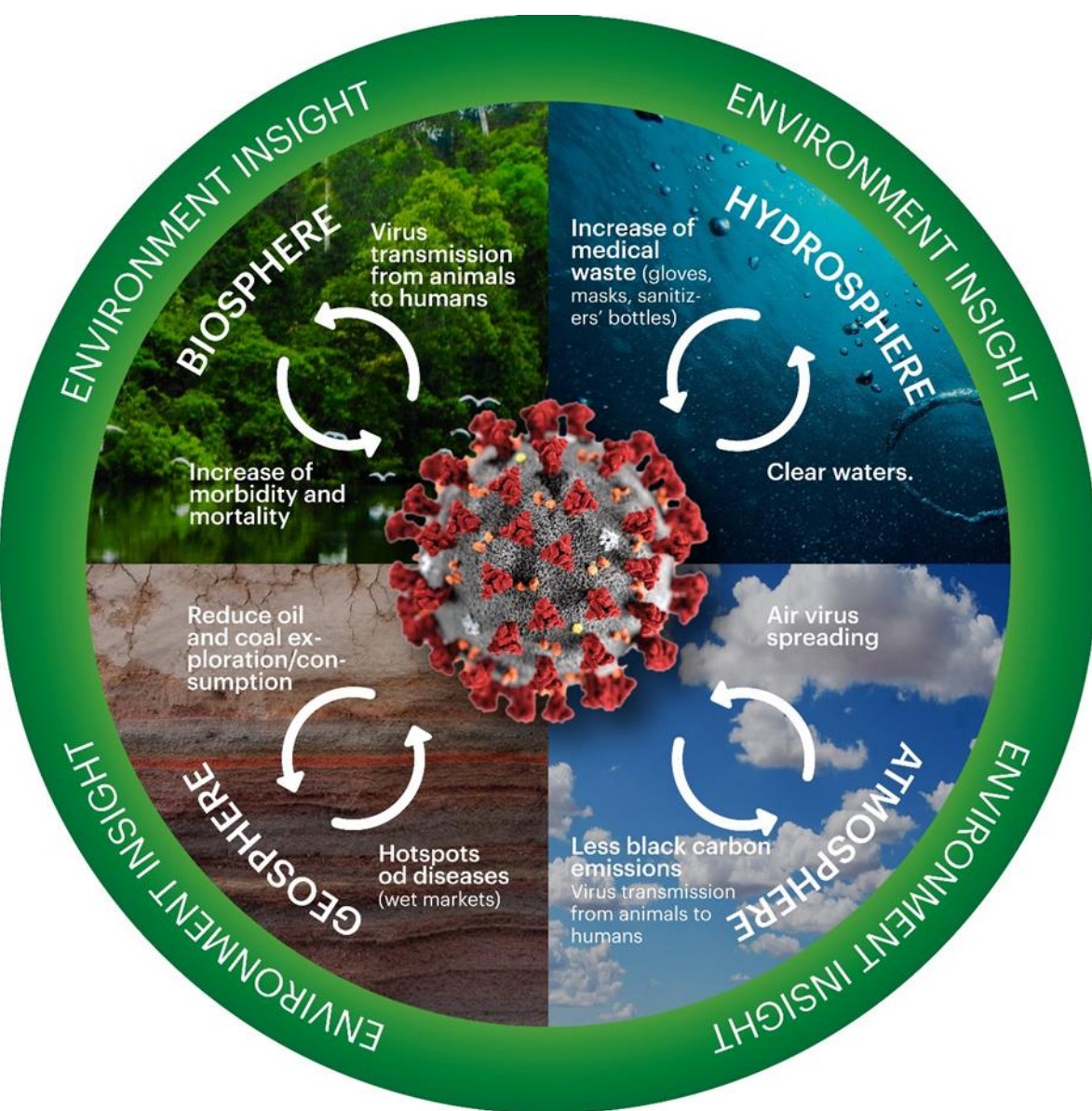

**Figure 2.** The integration of environmental insight and geoethics for a sustainable Earth sciences education.

## 5. Conclusions

The Earth system is being progressively damaged by human activity. Educating for sustainability means the integration of three pillars—environmental, social, and economic—in as many disciplines as possible. From science to literature to arts, education for sustainability should become an integral part of our education, our daily actions, and our routines. Although the Anthropocene has not yet been approved by the geological community as an epoch, it is clear that we humans are "a geological force" due to our overall impact on the Earth system. This Earth system analysis of the Anthropocene demands that environmental insight and Earth system thinking reach out of schools and out of books and into the public consciousness. Earth science education can help transform the current uninformed society into a society that more regularly, thoughtfully, and deliberately acts to support sustainability. Earth sciences must prepare students to become participative citizens and leaders of change. To embrace this goal, it is imperative to develop students' environmental insight in schools and universities, ensuring that students develop the high order competencies required to preserve the sustainability of life on Earth. The overwhelming complexity of the Earth system advocates a new, global approach to address its function and to solve urgent

global priorities for science, education, and policy. Environmental insight and individual geoethical behaviours are a necessary condition to continue towards the achievement of an Earth system sustainable governance.

**Author Contributions:** Conceptualization and investigation, N.O. and C.V.; writing—original draft preparation, C.V.; visualization, C.V.; writing—review and editing, N.O. and C.V.; supervision, N.O. All authors have read and agreed to the published version of the manuscript.

**Funding:** This research received no external funding.

**Institutional Review Board Statement:** The study did not require ethical approval.

**Informed Consent Statement:** Not applicable.

**Data Availability Statement:** No datasets were used in the present research.

**Conflicts of Interest:** The authors declare no conflict of interest.

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
