# Peer review of "Earth Science Education as a Key Component of Education for Sustainability"

_sustainability, doi:10.3390/su13031316_

Round 1

Reviewer 1 Report

Specific examples in the Geoethics section particularly but also Parts 2 and 3 be given slight more substance with examples or modeling strategies for K-16 educators as these constructs relate to sustainability.

Author Response

Point 1: Specific examples in the Geoethics section particularly but also Parts 2 and 3 be given slight more substance with examples or modelling strategies for K-16 educators as these constructs relate to sustainability.

Response 1: As far as the authors know there aren´t many activities validated to teach Geoethics in the classroom. Nevertheless, the handbook mentioned in the article (reference 60) has lots of Geoethics examples to apply in the classroom (please see line 250 and 251). There are some modelling activities related to sustainability although we find no reason to included them and no other kind of activities (field trips, lab activities, role-plays …). As the manuscript is not focus in geoscience activities in the classroom our option in to include none.

Reviewer 2 Report

The authors provide a well-structured overview of the field but do not provide any new research-based data.

Hence the scientific contribution to the field of sustainable education is low/moderate.

Author Response

Point 1: The authors provide a well-structured overview of the field but do not provide any new research-based data.

Hence the scientific contribution to the field of sustainable education is low/moderate.

Response 1: Unfortunately, Reviewer 2 missed the first line of the document.

In the line above the title of the manuscript appears the definition of the manuscript – "Review Paper".

We believe that if Reviewer 2 did not have missed the first line, then he/she would not base his/her negative criticism on the claimant that the manuscript "do not provide any new research-based data".

We believe that if he/she did not miss this line, then he/she would not expect for a new data, since this is the meaning of a review paper.

A review paper, exactly as it sounds, REVIEWs existing research-based publications and other types of articles, analyze them and suggest new insights following the analysis.

...And, this is exactly what we did in this manuscript. We reviewed published research-based studies and following this analysis, we came to a novel paradigm for education for sustainability. This manuscript is the first ever suggestion (following research-based studies) to base the education for sustainability on the Earth systems approach and the environmental insight.

If Reviewer 2 is familiar with any previous publications that have already suggested such approach for sustainability education, then we accept and respect his/her ruling. However, without any databased argumentation, such a strong rejection is unjustified and cannot be accepted.   

Reviewer 3 Report

The article has a strong message and highlights the importance of the Earth science education. The authors apply literature review/desktop research as their research method.
The authors provide a solid base to introduce the urge and need for higher intensity of human focus on “the persistent degradation if environmental quality”. They discuss the terminology, Earth system governance, Environment education, Education for sustainability but on the other hand, they do not introduce the information on prior experience form the education setting (e.g. resulting in the change of students’ attitudes and motivation) even though they mention “ the decade of education for sustainable development” and slow adaptation of the educational systems.

The articles e.g. 24-25 and 56 the authors refer to also discuss the issue in a way of theoretical description rather than bringing the data from the practice or their focus did not directly correspond to the authors' target. E.g. [24] is an interesting and significant study mapping the research 1970-2016– “The objective of this paper is to analyse the scientific production carried out in Portuguese universities in the fields of EE and ESD, by looking at the doctoral theses presented in public universities as a reference hallmark.“[25] is a descriptive article…
Adding info about e.g. action research or experiments conducted in the field would, I believe, make the text more valuable and the important message authors formulate stronger.

Line 161 reference ...according to the same author..... (the preceding source [1] and the next one [39] are not written by the same author (maybe 36?)

Author Response

Point 1: They discuss the terminology, Earth system governance, Environment education, Education for sustainability but on the other hand, they do not introduce the information on prior experience form the education setting (e.g. resulting in the change of students’ attitudes and motivation) even though they mention “ the decade of education for sustainable development” and slow adaptation of the educational systems. Adding info about e.g. action research or experiments conducted in the field would, I believe, make the text more valuable and the important message authors formulate stronger.

Response 1: Authors included some references to works undertaken in sustainability education and some achievements on lines 71 to 78 and references final list.

Formal sustainability education programs devoted to include rigorous systems thinking, social interactions and green principles have emerged in the last decade, and its success is generating initiatives to change the curricula [11]. From developing competences in primary pupils to engage higher education students, sustainability projects also demonstrate that integral approaches are urgently needed [12]. Research points that teaching methods providing a good introduction, supportive guidelines and include active participation and interactivity are the ones that integrate the most effective strategies for developing learners’ understanding, thinking and ability to act for sustainability [13, 14].

Jensen,C.; Kotaish, M.; Chopra, A.; Jacob, K.; Widekar, T.; Alam, R. Piloting a Methodology for Sustainability Education: Project Examples and Exploratory Action Research Highlights Emerging Science Journal  2019, 3,5, pp. 312-326.

Wamsler, C. Education for sustainability: Fostering a more conscious society and transformation towards sustainability International Journal of Sustainability in Higher Education 2020, 21,1,pp.112-130.

Jeronen, E.; Palmberg, I; Yli-Panula, E. Teaching Methods in Biology Education and Sustainability Education Including Outdoor Education for Promoting Sustainability—A Literature Review. Educ. Science 2017, 7,1.

Evans N. & Ferreira, J. What does the research evidence base tell us about the use and impact of sustainability pedagogies in initial teacher education? Environmental Education Research 2020, 26,1, pp.27-42.

Point 2: Line 161 reference ...according to the same author... (the preceding source [1] and the next one [39] are not written by the same author (maybe 36?)

Response 2:  Corrected and highlighted in red in the references section - reference 39 and 40 where swapped in first versions and reference 42 and 45 were the same. After changes having been made to answer the reviewers and correcting mistakes, those references are updated and can be found between 40 to 45 in the final reference list.

Round 2

Reviewer 3 Report

Authors reflected the comments and partly extended the oart about the previous research conducted in the field of education.